# Phenotypic and genomic characterization of ST11-K1 CR-hvKP with highly homologous $bla_{KPC-2}$-bearing plasmids in China

Yu-Ling Han,[1,2] Hua Wang,[1,3] Hong-Zhe Zhu,[1,2] Ying-Ying Lv,[1] Wen Zhao,[1] Yan-Yan Wang,[1] Jian-Xun Wen,[4] Zhi-De Hu,[1] Jun-Rui Wang,[1] Wen-Qi Zheng[1,2]

**ABSTRACT** Carbapenem-resistant hypervirulent *Klebsiella pneumoniae* (CR-hvKP) strains present a significant global public health threat due to their high mortality rates. This study investigated the genomic characteristics of seven ST11-K1 CR-hvKP isolates harboring highly homologous KPC-2-encoding multidrug-resistance plasmids. The strains were isolated from a Chinese tertiary hospital between 2017 and 2020. Whole-genome sequencing and bioinformatic analysis revealed various antibiotic resistance genes (ARGs) and virulence determinants. The $bla_{KPC-2}$-bearing plasmids that contain multiple antibiotic-resistance genes were also identified in these strains. ISfinder and Orifinder were applied to identify insertion sequences (IS) and conjugation-related factors among these $bla_{KPC-2}$-bearing plasmids. The $bla_{KPC-2}$ was highly consistent in seven $bla_{KPC-2}$-bearing plasmids (IS*Kpn6*-$bla_{KPC-2}$-IS*Kpn27*-IS*Yps3*-IS*26*). In addition, we found a region composed of IS*IR*, Tn*5393*, and IS*26*. It was located upstream of the $bla_{CTX-M-15}$ gene and presented in six $bla_{KPC-2}$-bearing plasmids, with pCR-hvKP221-KPC-P3 as an exception. Conjugation experiments demonstrated the horizontal transfer of resistance plasmids pCR-hvKP128-KPC-P1 and pCR-hvKP132-KPC-P1 across species. Notably, pLVPK-like virulence plasmids carrying virulence gene clusters pCR-hvKP173-Vir-P1, and pCR-hvKP221-Vir-P1 were also detected. A fusional plasmid pCR-hvKP221-Vir-P2, which carries virulence gene clusters and ARGs, was also identified. Five CR-hvKP strains displayed enhanced biofilm formation and high virulence *in vivo* infection models. Phylogenetic and single nucleotide polymorphism (SNP) analyses indicated a close genetic relationship among the isolates, suggesting a subclade. These findings highlight the complex genetic profiles and potential transmission mechanisms of CR-hvKP strains.

**IMPORTANCE** We reported seven CR-hvKP strains all carried a highly homologous $bla_{KPC-2}$ integrated IncFⅡ-resistant plasmid, and two strains harbored virulence plasmids. Conjugation experiments confirmed the transferability of these plasmids, indicating a potential for resistance spread. Phylogenetic analysis clarified the relationship among the CR-hvKP isolates. This study provides insights into the phenotypic and genomic characteristics of seven ST11-K1 CR-hvKP strains. The high prevalence and potential for local outbreaks emphasize the need for effective control measures.

**KEYWORDS** carbapenem-resistant hypervirulent *Klebsiella pneumoniae*, $bla_{KPC-2}$, multidrug resistance, plasmid, whole-genome sequence

*K*lebsiella pneumoniae (*K. pneumoniae*) is a Gram-negative bacterium that commonly causes hospital-acquired infections, especially in immunocompromised patients (1). It can be categorized into classic *K. pneumoniae* (cKP) and hypervirulent *K. pneumoniae* (hvKP) based on their virulence characteristics (2). HvKP usually causes invasive infections such as endophthalmitis, liver abscesses, pneumonia, bacteremia, and septic shock (3, 4). HvKP strains can acquire multidrug resistance (MDR) by acquiring multiple

**Peer Reviewer** Piklu Roy Chowdhury, University of Technology Sydney, Sydney, Australia

Address correspondence to Wen-Qi Zheng, zhengwenqi2011@163.com.

Yu-Ling Han and Hua Wang contributed equally to this article. Author order was determined by the corresponding author after negotiation.

The authors declare no conflict of interest.

See the funding table on p. 14.

antibiotic-resistance genes. Meanwhile, MDR cKP strains can also acquire virulence genes, resulting in the emergence of MDR-hvKP strains (5–8). Carbapenem resistance, often accompanied by resistance to other antimicrobial classes, poses a significant public health threat. The World Health Organization (WHO) has categorized carbapenem-resistant *Enterobacterales* (CRE) as the most urgent priority due to limited treatment options (9, 10). In China, carbapenem-resistant *K. pneumoniae* (CRKP) strains, accounting for approximately 90% of CRE cases, frequently cause difficult-to-treat infections (11). Furthermore, carbapenem-resistant hypervirulent *K. pneumoniae* (CR-hvKP), which combines carbapenem resistance and hypervirulence, has become a global concern since 2015 due to its high mortality and morbidity rates (5, 12, 13).

Despite the high prevalence and worse outcomes of CR-hvKP, the therapeutic options for treating this infection are limited (14). According to the published studies, China is mainly affected by the CR-hvKP (15–18), with sequence type 11 (ST11) and KPC carbapenemase-producing CR-hvKP strains being predominant (19–21). The acquisition of carbapenemase-producing plasmids by hvKP strains is a major factor contributing to the evolution and spread of CR-hvKP strains (6, 12). In addition, CR-hvKP strains can also be formed when hvKP strains acquire conjugative KPC-producing plasmids, or CRKP strains acquire nonconjugative virulence plasmids (12). Therefore, conjugative KPC-producing plasmids in CR-hvKP strains promote MDR and facilitate the spread of virulence plasmids to other CRKP strains (12). Moreover, the carbapenemase-producing plasmids, especially KPC-positive plasmids, have a high potential for inter-strain spread (22). The wider spread of CR-hvKP strains is thus facilitated when hvKP strains acquire KPC-positive plasmids. However, there is limited information on the long-term prevalence of KPC-positive plasmids circulating among CR-hvKP strains in a specific region. This study reports seven ST11 CR-hvKP clinical strains from a tertiary hospital in China between 2017 and 2020. We analyzed the genotypic, phenotypic, virulence, and phylogenetic characteristics of these CR-hvKP strains. Our findings reveal not only multi-antimicrobial resistance, strong biofilm formation ability, and high virulence characteristics but also the presence of seven highly homologous KPC-2-encoding multidrug-resistant plasmids among the seven ST11 CR-hvKP strains isolated over 4 years, indicating the long-term prevalence of these plasmids.

## RESULTS

### Clinical characteristics of seven CR-hvKP isolates

In our previous study, 124 CRKP strains were collected from the Affiliated Hospital of Inner Mongolia Medical University between August 2015 and October 2020 (23). Seven of these strains exhibited hypermucoviscous phenotypes, which were confirmed using the string test. These seven CRKP isolates were thus defined as CR-hvKP. They were isolated from seven patients aged 1–77 years between 2017 and 2020. These strains were isolated from blood (*n* = 3), sputum (*n* = 2), and urine (*n* = 2). Table 1 presents a summary of the clinical information for the CR-hvKP strains.

### Antimicrobial susceptibility profiles

The antimicrobial susceptibility test results for the seven CR-hvKP isolates are presented in Table 2. All isolates demonstrated high-level resistance to aztreonam, amikacin, meropenem, ertapenem, ceftriaxone, cefotaxime, imipenem, ciprofloxacin, and ceftazidime. Notably, most antibiotics' minimal inhibitory concentrations (MICs) were over 128 µg/mL. In addition, some isolates displayed resistance to gentamicin (6/7), tobramycin (5/7), and chloramphenicol (5/7). However, six isolates were susceptible to tigecycline, and all strains remained susceptible to ceftazidime/avibactam and showed intermediate susceptibility to fosfomycin and polymyxin B.

TABLE 1 Summary of clinical information for seven CR-hvKP strains[a]

| Isolates | Age (years)/sex | Isolation date (month/year) | Specimen | Ward | Primary diagnosis |
|---|---|---|---|---|---|
| CR-hvKP005 | 74/M | 03/2017 | Blood | Surgical Department | Postoperative hilar cholangiocarcinoma |
| CR-hvKP006 | 29/M | 10/2020 | Sputum | Rehabilitation Department | Recovery phase of traumatic brain injury |
| CR-hvKP26 | 1/M | 03/2017 | Sputum | Pediatric Intensive Care Unit | Hirschsprung's disease |
| CR-hvKP128 | 68/M | 09/2019 | Blood | Surgical Department | Postoperative hilar cholangiocarcinoma |
| CR-hvKP132 | 77/F | 04/2019 | Urine | Traditional Medicine Department | NA |
| CR-hvKP173 | 44/M | 06/2018 | Blood | Rehabilitation Department | Postoperative cervical spine fracture |
| CR-hvKP221 | 71/M | 07/2020 | Urine | Respiratory Department | Sepsis |

[a]M, male; F, female; NA, not applicable.

## Genomic characteristics of seven CR-hvKP strains

Whole-genome sequencing (WGS) was performed to explore the genomic characteristics of seven CR-hvKP strains. Genome assembly statistics and assembly plasmids alignment results are shown in Tables S1 and S2, respectively. WGS analysis revealed that all the CR-hvKP strains contained a chromosome and at least two plasmids. Molecular typing identified all isolates belonging to ST11, a clonal group known as CG258, and the K1 serotype. Table 3 summarizes the molecular characteristics of the seven CR-hvKP strains. All plasmids carrying the $bla_{KPC-2}$ gene belong to the IncFII/IncR type. The replicon types of pCR-hvKP173-Vir-P1 and pCR-hvKP221-Vir-P1 were IncHI1B/repB. The replicon type of pCR-hvKP221-Vir-P2 was IncFIB/IncFIC. In addition, all of the detected plasmids were found to possess one or more conjugal transfer elements, including relaxase, origin of transfer (oriT), type IV coupling proteins (T4CP), and type IV secretion system (T4SS) (Table 3).

All seven CR-hvKP isolates carried 19 antibiotic resistance genes on chromosomes, including $bla_{SHV-11}$, $tetA/B$, $sul4$, $oqxB$, $mcr-8$, $fosA$, $mupB$, etc (Fig. 1A). In addition, four strains carried $oqxB$; three strains carry $sul1$ and $aadA2$. The resistance genes $bla_{KPC-2}$, $bla_{SHV-12}$, $bla_{CTX-M-65}$, $bla_{TEM-1}$, and $rmtB$ were located on the plasmid of the CR-hvKP strain. Seven strains carried $bla_{KPC-2}$ genes, six strains carried $bla_{SHV-12}$ and $bla_{CTX-M-65}$ genes, and five strains carried $bla_{TEM-1}$ and $rmtB$ genes (Fig. 1A). Interestingly, the efflux pump-related gene $oqxB$ was found on both the chromosomes and plasmid of CR-hvKP173, and the sulfonamide resistance gene $sul1$ was present on both the chromosomes and plasmid of CR-hvKP221 (Fig. 1A). Notably, we found $ompK37$ mutant

TABLE 2 Antimicrobial susceptibility results for seven CR-hvKP strains[a]

| Antibiotics | CR-hvKP005 | CR-hvKP006 | CR-hvKP26 | CR-hvKP128 | CR-hvKP132 | CR-hvKP173 | CR-hvKP221 |
|---|---|---|---|---|---|---|---|
| Aztreonam | >128/R | >128/R | >128/R | >128/R | >128/R | >128/R | >128/R |
| Amikacin | >128/R | >128/R | >128/R | >128/R | >128/R | >128/R | >128/R |
| Meropenem | >128/R | >128/R | >128/R | >128/R | >128/R | >128/R | >128/R |
| Ertapenem | >128/R | >128/R | >128/R | >128/R | >128/R | >128/R | >128/R |
| Ceftriaxone | >128/R | >128/R | >128/R | >128/R | >128/R | >128/R | >128/R |
| Cefotaxime | >28/R | >128/R | >128/R | >128/R | >128/R | >128/R | >128/R |
| Imipenem | 64/R | >128/R | 16/R | >128/R | 64/R | 32/R | 32/R |
| Ciprofloxacin | 32/R | 32/R | 32/R | 32/R | 16/R | 16/R | 16/R |
| Ceftazidime | 64/R | 128/R | 32/R | 16/R | 16/R | 64/R | 32/R |
| Gentamicin | 16/R | 1/S | >128/R | >128/R | >128/R | >128/R | >128/R |
| Tobramycin | 1/S | 2/S | >128/R | >128/R | >128/R | >128/R | >128/R |
| Chloramphenicol | 4/S | >128/R | >128/R | 6/I | 64/R | >128/R | 32/R |
| Fosfomycin | >128/I | >128/I | >128/I | >128/I | >128/I | >128/I | >128/I |
| Polymyxin B | 2/I | 2/I | 2/I | 0.5/I | 1/I | 1/I | 1/I |
| Tigecycline | 1/S | 1/S | 2/S | 2/S | 2/S | 4/I | ≤0.5/S |
| Ceftazidime/avibactam | 4/S | 4/S | 8/S | 2/S | 4/S | 1/S | 4/S |

[a]R, resistant; I, intermediary; S, susceptible. The interpretation criteria of antimicrobial susceptibility results: except for tigecycline, the remaining 15 antibiotics were judged according to the Clinical and Laboratory Standards Institute (CLSI) M100 document; the susceptibility results of tigecycline were interpreted according to the U.S. Food and Drug Administration (FDA) standard.

**TABLE 3** Molecular characterization of the chromosome and resistant and virulent plasmids of the CR-hvKP isolates[a]

| Isolates | Genome | Replicon | Size (bp) | GC (%) | Conjugal transfer elements | | | | Gene | | Accession number |
|---|---|---|---|---|---|---|---|---|---|---|---|
| | | | | | Relaxase | oriT[b] | T4CP[c] | T4SS[d] | ARG[e] | VF[f] | |
| CR-hvKP005 | Chromosome | NA[g] | 5,496,361 | 57.35 | ✓ | × | × | ✓ | ✓ | ✓ | CP119013 |
| | pCR-hvKP005-KPC-P1 | IncFII, IncR | 69,455 | 54.30 | ✓ | × | × | × | ✓ | × | CP119014 |
| CR-hvKP006 | Chromosome | NA | 5,467,613 | 57.36 | ✓ | × | × | ✓ | ✓ | ✓ | CP119017 |
| | pCR-hvKP006-KPC-P1 | IncFII, IncR | 111,731 | 55.27 | × | × | × | ✓ | ✓ | × | CP119018 |
| CR-hvKP26 | Chromosome | NA | 5,422,087 | 57.40 | ✓ | × | × | ✓ | ✓ | ✓ | CP119021 |
| | pCR-hvKP26-KPC-P1 | IncFII | 89,906 | 53.87 | × | × | × | ✓ | ✓ | × | CP119022 |
| CR-hvKP128 | Chromosome | NA | 5,453,906 | 57.40 | ✓ | × | × | ✓ | ✓ | ✓ | CP119026 |
| | pCR-hvKP128-KPC-P1 | IncFII, IncR | 154,719 | 53.08 | × | ✓ | × | ✓ | ✓ | × | CP119027 |
| CR-hvKP132 | Chromosome | NA | 5,533,017 | 57.33 | ✓ | × | × | ✓ | ✓ | ✓ | CP119029 |
| | pCR-hvKP132-KPC-P1 | IncFII, IncR | 154,719 | 53.08 | × | ✓ | × | ✓ | ✓ | × | CP119030 |
| CR-hvKP173 | Chromosome | NA | 5,464,083 | 57.35 | ✓ | × | × | ✓ | ✓ | ✓ | CP119034 |
| | pCR-hvKP173-Vir-P1 | IncHI1B, repB | 178,873 | 49.37 | × | ✓ | ✓ | × | × | ✓ | CP119035 |
| | pCR-hvKP173-KPC-P2 | IncFII, IncR | 144,880 | 53.49 | × | ✓ | × | ✓ | ✓ | × | CP119036 |
| CR-hvKP221 | Chromosome | NA | 5,517,123 | 57.26 | ✓ | × | × | ✓ | ✓ | ✓ | CP119040 |
| | pCR-hvKP221-Vir-P1 | IncHI1B, repB | 145,168 | 50.30 | × | × | × | × | × | ✓ | CP119041 |
| | pCR-hvKP221-Vir-P2 | IncFIB, IncFIC | 140,816 | 50.68 | × | ✓ | × | ✓ | ✓ | ✓ | CP119042 |
| | pCR-hvKP221-KPC-P3 | IncFII, IncR | 136,943 | 53.02 | ✓ | × | × | ✓ | ✓ | × | CP119043 |

[a]The pattern "×" represents that the target element/gene was not predicted; the pattern "✓" represents that the target element/gene was predicted. The accession number was obtained after the WGS data of seven CR-hvKP strains in this study were uploaded to NCBI.
[b]oriT: the origin of transfer.
[c]T4CP: IV coupling proteins.
[d]T4SS: type IV secretion system.
[e]ARG: an antibiotic resistance gene.
[f]VF: virulence factor gene.
[g]NA: not applicable.

changed from ATT to ATG, and from AAC to GGC on the chromosomes of seven CR-hvKP strains, which had a certain correlation with carbapenems resistance (24). In addition, a mutant of *ompK36* (the nucleotide site from AAC to AGC, CTT to GTA, ACC to CCA, and AAC to GAA) was detected in CR-hvKP221, which could lead to a cephalosporin resistance phenotype (25).

The virulence genes *entA/B/C/D/E/F/S*, *ybtA/E/P/Q/U/S/T/X*, *iroE/N*, *wzc*, *fimA/B/C/D/E/F/G/H/I/K*, and *ibeB* were detected on the chromosomes of seven CR-hvKP strains (Fig. 1B). Notably, The *rmpA* gene was found on the chromosomes of six CR-hvKP strains except CR-hvKP173. CR-hvKP173 carried one virulence plasmid, while CR-hvKP221 carried two virulence plasmids. On the plasmid of CR-hvKP173, the *iucA/B/C/D* and *rmpA* genes were detected, and on the plasmid of CR-hvKP221, genes *iroB/C/D*, *sitD*, and *iucA/B/C/D* were present (Fig. 1B). The functions of all detected resistance and virulence genes are summarized in Table S3.

## Genetic features of the IncFII type *bla*$_{KPC-2}$-harboring plasmid

As shown in Fig. 2, all CR-hvKP isolates harbored a *bla*$_{KPC-2}$-bearing plasmid, namely, pCR-hvKP005-KPC-P1, pCR-hvKP006-KPC-P1, pCR-hvKP26-KPC-P1, pCR-hvKP128-KPC-P1, pCR-hvKP132-KPC-P1, pCR-hvKP173-KPC-P2, and pCR-hvKP221-KPC-P3, respectively. The length of these seven *bla*$_{KPC-2}$-bearing plasmids was 69,455, 111,731, 89,906, 154,719, 154,719, 144,880, and 136,943 bp respectively. Their GC contents range from 49.37% to 64.75% (Table 3). In addition, they contained various resistance genes such as *bla*$_{KPC-2}$, *bla*$_{SHV-12}$, *bla*$_{CTX-M-65}$, *bla*$_{TEM-1}$, *rmtB*, *catA2*, and/or *fosA*. We have aligned the sequences of the *bla*$_{KPC-2}$-bearing plasmids. We found that the query coverage ranged from 48% to 100%, and the percent identities were 99% (Fig. 2A and B). Notably, the sequences of plasmids pCR-hvKP128-KPC-P1 and pCR-hvKP132-KPC-P1 were completely identical, exhibiting 100% coverage and identity. (Fig. 2B).

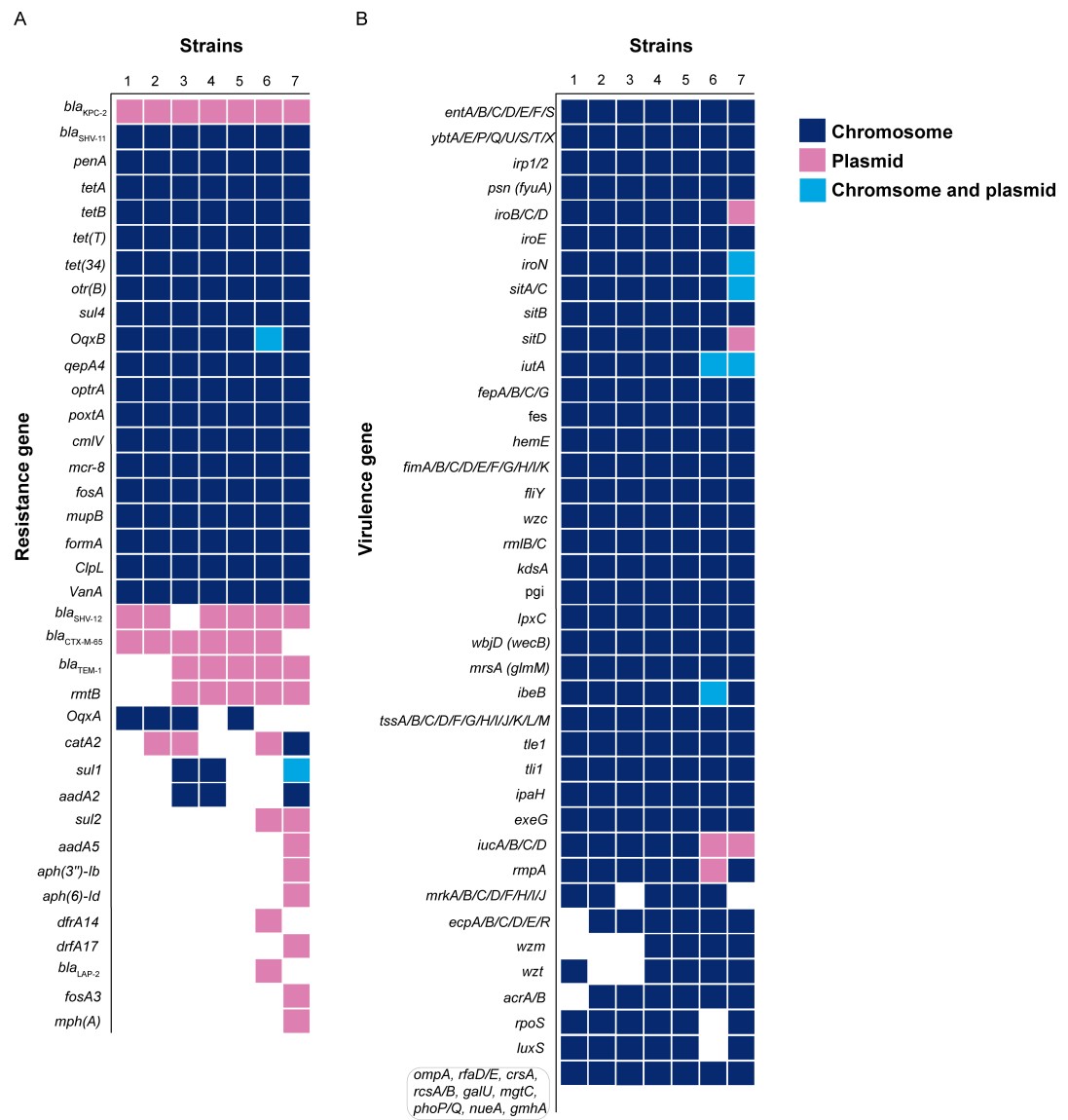

**FIG 1** The resistance genes (A) and virulence genes (B) of seven CR-hvKP strains. The numbers 1–7 represent CR-hvKP005, CR-hvKP006, CR-hvKP26, CR-hvKP128, CR-hvKP132, CR-hvKP173, and CR-hvKP221, respectively.

As shown in Fig. 2C, the genetic context of the $bla_{KPC-2}$ gene (IS$Kpn6$-$bla_{KPC-2}$-IS$Kpn27$-IS$Yps3$-IS$26$) was identical to seven $bla_{KPC-2}$-bearing plasmids. In addition, except pCR-hvKP221-KPC-P3, a region consisting of Tn$5393$, and IS$26$, and located upstream of the $bla_{CTX-M-15}$ was detected in six $bla_{KPC-2}$-bearing plasmids. Furthermore, the genetic context of the $bla_{TEM-1}$ in pCR-hvKP173-KPC-P2 (IS$Sbo1$-IS$26$-Tn$5393$-$bla_{TEM-1}$-$rmtB$-IS$15DI$) was nearly identical to that of pCR-hvKP221-KPC-P3, although the latter plasmid lacked the $bla_{CTX-M-15}$. In contrast to pCR-hvKP173-KPC-P2, the pCR-hvKP221-KPC-P3 plasmid contained an additional resistance gene, $fosA3$, along with a DNA fragment upstream of $bla_{TEM-1}$.

## Sequence analysis of the virulence plasmid

WGS analysis revealed that only CR-hvKP173 and CR-hvKP221 strains carried virulence plasmids. The CR-hvKP173 strain harbored a 178,873 bp virulence plasmid named pCR-hvKP173-Vir-P1 (Fig. 3). This plasmid contained the multidrug efflux pump gene $oqxB$, as well as virulence genes ($iucA/B/C/D$, $iutA$, and $rmpA$), and the copper/sliver resistance

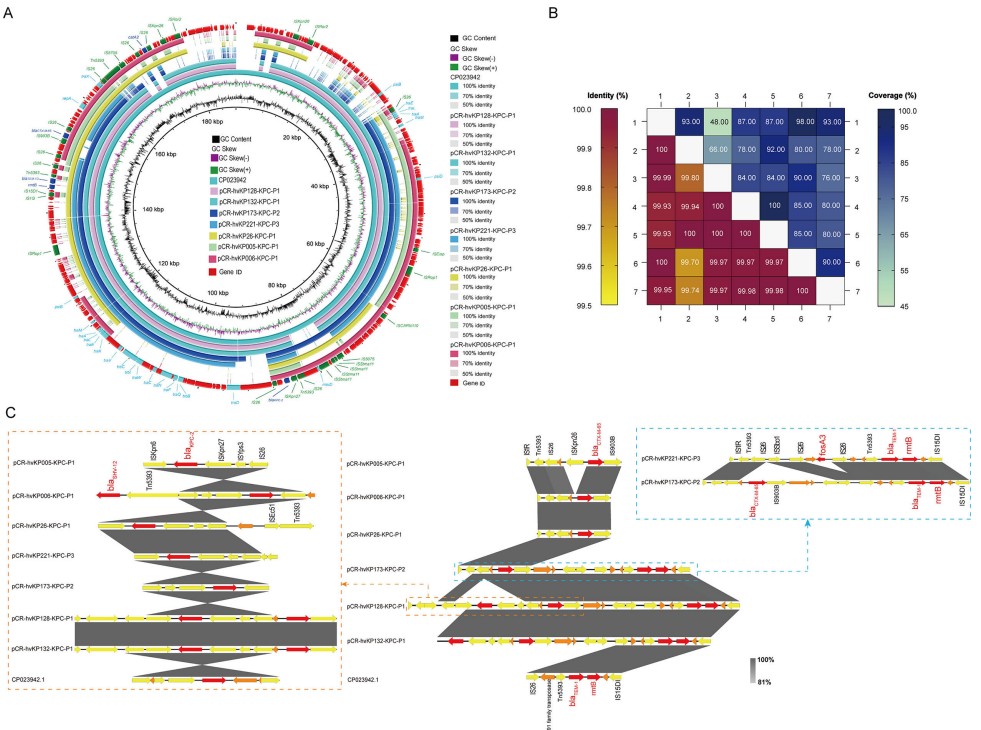

**FIG 2** Alignments of seven *bla*$_{KPC-2}$-bearing plasmids. (A) The circle graphs of seven *bla*$_{KPC-2}$-bearing plasmids were drawn by BRIG. (B) BLAST analysis of seven *bla*$_{KPC-2}$-bearing plasmids in pairs in the NCBI nucleotide database. The value represents coverage and identity. The numbers 1–7 represent CR-hvKP005, CR-hvKP006, CR-hvKP26, CR-hvKP128, CR-hvKP132, CR-hvKP173, and CR-hvKP221, respectively. (C) Collinear diagram of seven *bla*$_{KPC-2}$-bearing plasmids by Easyfig. Resistance genes are indicated in red, IS elements in yellow, and all other ORFs in orange. Regions sharing identical or near-identical sequences across plasmids are indicated by the gray shading among the representations of different plasmids. The reference plasmid was a plasmid of *K. pneumoniae* (GenBank No. CP023942.1).

gene *ibeB*. Comparative genomics demonstrated that pCR-hvKP173-Vir-P1 exhibited high similarity to several reference hypervirulence plasmids, including pLVPK (Accession: NC_005249.1), pVir-CR-HvKP4 (Accession: NZ_MF437313.1), pVir-CR-hvKP-C789 (Accession: NZ_CP034416.1), and pK2044 (Accession: NC_006625.1), which isolated from hvKP strains. Notably, pVir-CR-hvKP-C789 and pVir-CR-HvKP4 were isolated from ST11 KPC-2-producing CR-hvKP strains in China (26, 27). These findings suggest a potential common evolutionary origin between the virulence plasmids identified in this study and those found in other *K. pneumoniae* strains.

The CR-hvKP221 strain was found to carry two virulence plasmids: pCR-hvKP221-Vir-P1 and pCR-hvKP221-Vir-P2. The pCR-hvKP221-Vir-P1 has a size of 145,168 bp and an average GC content of 50.30%. It belongs to the IncFIB/IncFIC (FII) plasmid type and contains the virulence genes *iucA/B/C/D* and *iutA*. Comparative genomics analysis revealed that pCR-hvKP221-Vir-P1 exhibited the highest similarity with 99% query coverage and 100% identity to the CRKP78R plasmid p2 (Accession: NZ_CP066255.1), which is the size of 215,966 bp and was isolated from an ST11 CRKP strain isolated in China (28). In addition, pCR-hvKP221-Vir-P1 displayed 99.86% similarity (with 88% query coverage) to pLVPK (Accession: NC_005249.1) and 99.83% similarity (with 74% query coverage) to plasmid pVir-CR-HvKP4 (Accession: NZ_MF437313.1), as shown in Fig. 4A.

The size of pCR-hvKP221-Vir-P2 was determined to be 140,816 bp, with an average GC content of 50.68%. As illustrated in Fig. 4B, pCR-hvKP221-Vir-P2 consists of two regions enriched with virulence genes, *iroB/C/D/N,* and *sitA/C/D-iucA/B/C-iutA*. In addition, it contains a T4SS region and an antibiotic resistance genes (ARGs) enrichment region, which possess multi-replicon IncFIB and IncFIC plasmids. The T4SS region harbors transfer-related genes such as *traX* and *traM*. By contrast, the ARG enrichment

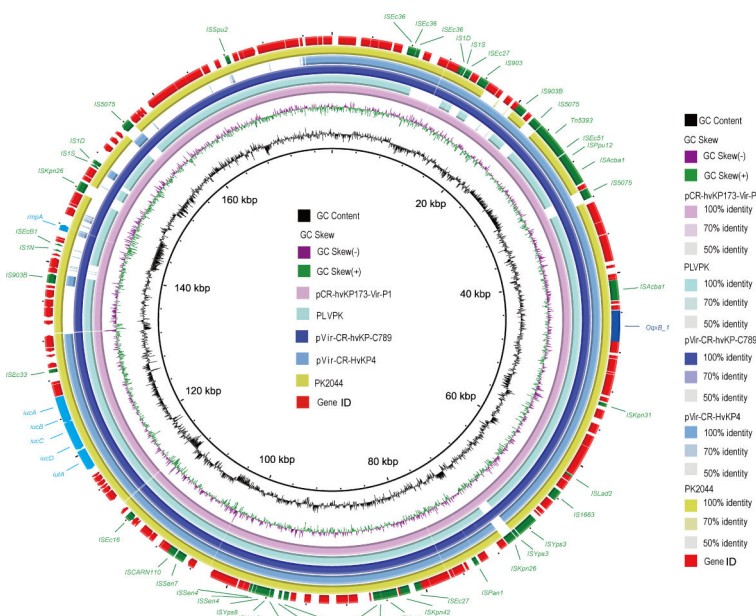

**FIG 3** A circular map created by BRIG depicts the similarity of alignments between pCR-hvKP173-Vir-P1 and similar plasmids reported in previous studies.

region carries *tet(A)*, *aph(6)-ld, aadA5*, *aph(3″)-lb*, *sul1/2*, *mph(A)*, and *dfrA17*, conferring resistance to tetracycline, aminoglycosides, sulfonamides, macrolides, trimethoprim, and β-lactam antibiotics (Fig. 4B). A BLAST alignment revealed that pCR-hvKP221-Vir-P2 exhibited the highest similarity to *Escherichia coli* strain D6 plasmid A (Accession: CP010149.1), with 100% query coverage and 99.86% identity (Fig. 4B).

## Phylogenetic analysis of CR-hvKP

To investigate the phylogenetic relationship among the CR-hvKP strains, a WGS single-nucleotide polymorphism (wgSNP)-based phylogeny maximum likelihood (ML) tree was conducted using 7 CR-hvKP strains and 395 reference strains. This analysis utilized the Gubbins software, which is designed for SNP-based core genome phylogeny. Before tree construction, the 402 genomes involved were compared with the reference genome CR-hvKP221. The results of the SNP analysis and the genome coverage between CR-hvKP221 and the 402 strains are presented in Table S4. A total of 99,990 columns were used to construct the tree.

As shown in Fig. 5, the resulting phylogenetic tree had two main branches. Notably, all seven CR-hvKP strains clustered within one of these branches, suggesting a shared ancestry. It is important to highlight that all strains in this branch were isolated from China (Fig. 5). The CR-hvKP strains were further categorized into two distinct sub-branches. Specifically, CR-hvKP005, CR-hvKP128, CR-hvKP132, CR-hvKP173, and CR-hvKP221 clustered on one sub-branch (bootstrap value of 100), while CR-hvKP006 and CR-hvKP26 formed another sub-branch (also with a bootstrap value of 100). Interestingly, CR-hvKP128 and CR-hvKP132 emerged from the same sub-branch and exhibited a close relationship (bootstrap value of 99), indicating their recent common ancestry (Fig. 5).

## Conjugation of the *bla*$_{KPC-2}$-bearing plasmid

Conjugation experiments revealed that only the *bla*$_{KPC}$-positive plasmids from CR-hvKP128 and CR-hvKP132 strains were successfully transferred. These results suggest that the plasmids pCR-hvKP128-P1 and pCR-hvKP132-P1, which harbor the *bla*$_{KPC-2}$, are capable of interspecies transmission.

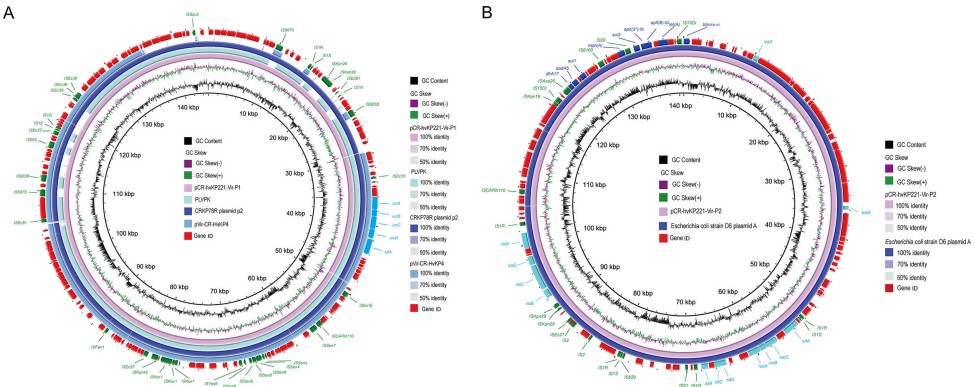

**FIG 4** Circular map created by BRIG depicts the similarity of alignments between plasmids reported in previous studies and pCR-hvKP221-Vir-P1 (A) and pCR-hvKP221-Vir-P2 (B).

## Biofilm formation assay *in vitro*

As shown in Fig. 6, the $OD_{600}$ value of these isolates ranged between 0.136 nm and 0.338 nm. Among the studied isolates, CR-hvKP005, CR-hvKP006, CR-hvKP132, CR-hvKP173, and CR-hvKP221 have strong biofilm formation capabilities, CR-hvKP26 and CR-hvKP128 displayed weak biofilm formation (Fig. 6). Notably, the biofilm formation ability of CR-hvKP005 and CR-hvKP221 was comparable to that of NTUH-K2044, a well-known hypervirulent strain.

## Virulence characteristics *in vivo*

In all, 106 virulence genes were detected by WGS analysis (Table S3), and 78 virulence genes were detected in all seven CR-hvKP strains (Fig. 1), of which siderophore-related virulence genes accounted for 38.5% (30/78), fimbriae-related genes accounted for 15.4% (12/78), and secretion system-related genes accounted for 20.5% (16/78). *In vivo* experiments showed that the virulence phenotype of seven CR-hvKP strains was consistent with the abundance of virulence genes. The virulence characteristics of the CR-hvKP strains were assessed using the *Galleria mellonella* (*G. mellonella*) infection model and a murine infection model. As depicted in Fig. 7A, the *G. mellonella* infection model demonstrated that all CR-hvKP clinical isolates exhibited a hypervirulent phenotype. Interestingly, CR-hvKP26 displayed a virulence level surpassed that of clinical isolate IM7 but comparable to that of the hypervirulent strain NTUH-K2044, while the virulence of CR-hvKP005, CR-hvKP006, CR-hvKP128, CR-hvKP132, CR-hvKP173, and CR-hvKP221 strains surpassed that of NTUH-K2044 and the clinical isolate IM7 (Fig. 7A).

The results of the murine infection experiments indicated that the virulence of CR-hvKP005 and CR-hvKP006 strains was similar to that of NTUH-K2044. In addition, CR-hvKP221, CR-hvKP173, and CR-hvKP132 demonstrated greater virulence compared to the clinical isolate IM7 (cKP control), whereas CR-hvKP26 and CR-hvKP128 exhibited similar virulence as IM7 (Fig. 7B).

## DISCUSSION

In recent years, the emergence of CR-hvKP strains has posed a significant global public health challenge. These strains have evolved from the ST11 CRKP strains, becoming more virulent and complex (6). This study aimed to investigate the resistance and hypervirulence phenotype, as well as genomic characteristics, of seven ST11 CR-hvKP strains. We identified two pLVPK-like plasmids and a fusion plasmid harboring ARGs and virulence genes in two CR-hvKP isolates. Notably, we found seven highly homologous MDR resistance plasmids containing the $bla_{KPC-2}$. Multiple insertion sequences—IS*Kpn6*, IS*Kpn27*, IS*Yps3*, and IS*26*—were found surrounding the $bla_{KPC-2}$. The presence of these plasmids complicates treatment options and heightens the risk of transmission

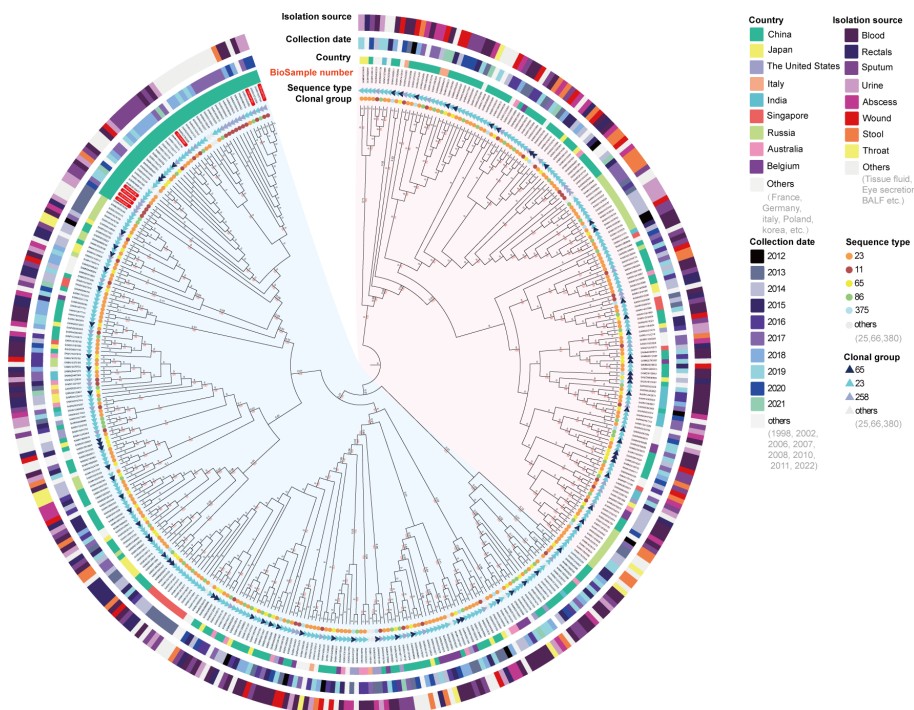

**FIG 5** Phylogenetic analysis of seven CR-hvKP strains and 395 reference strains using software MEGA. The cladogram tree was constructed using the maximum likelihood (ML) method. Blue and red backgrounds represent two major branches. The black numbers represent the branch length (rounded to two decimal places), and the red numbers represent the bootstrap values (only values between 80 and 100 are displayed).

of high-risk elements. Our findings underscore the significant role of these highly homologous *KPC-2*-encoding plasmids in the evolution of CR-hvKP, attributed to their extensive transmission potential and prolonged presence in healthcare settings.

ST11-K1 KPC-2-producing CR-hvKP is the most prevalent strain in China (6, 29, 30). Previous studies indicated that $bla_{KPC-2}$ and the ST11-K1 clones significantly contribute to the development of the hypervirulent phenotype and carbapenem resistance (20). Our findings support these findings because all seven CR-hvKP isolates were ST11 clones and could produce the KPC-2 carbapenemase. ST11 CR-hvKP strains carry an IncFII/IncR plasmid harboring the $bla_{KPC-2}$ gene, which facilitates the dissemination of their resistance phenotype in healthcare settings (31–33). Consistent with these findings, our study revealed that each CR-hvKP strain carried an IncFII/IncR type plasmid containing the $bla_{KPC-2}$, conferring multi-antibiotic resistance characteristics. Notably, these seven $bla_{KPC-2}$-carrying plasmids exhibited a high level of homology, although they were collected over 4 years. We investigated the horizontal transfer potential of the seven $bla_{KPC-2}$-carrying plasmids through conjugation assays and found that only pCR-hvKP128-KPC-P1 and pCR-hvKP132-KPC-P1 were successfully transferred to *Escherichia coli*. The oriTfinder analysis failed to identify a complete conjugative transfer region encompassing relaxase, oriT, T4CP, and T4SS on these seven plasmids. However, we found that pCR-hvKP128-KPC-P1 and pCR-hvKP132-KPC-P1 contained two proteins that shared high similarity to relaxase (BLASTp identities: 97.48%) and T4CP (BLASTp identities: 97.98%), respectively, suggesting that pCR-hvKP128-KPC-P1 and pCR-hvKP132-KPC-P1 possess conjugative transfer capabilities and can successfully transfer to *Escherichia coli*.

MDR plasmids are formed through recombination events that involve multiple mobile genetic elements (MGEs) (34). IS*26* acts as a major contributor to the formation of MDR regions (35–37) and is associated with the insertion and deletion of the resistance gene $bla_{KPC-2}$ (38–40). Furthermore, a study by Yang et al. found that IS*26* and IS*903B* could

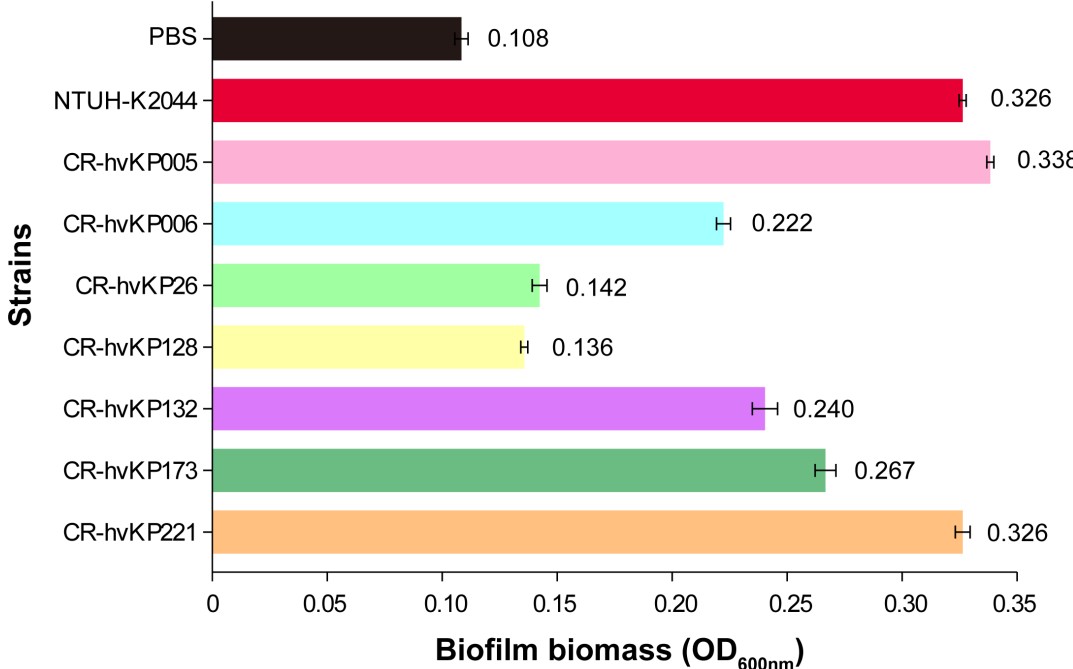

**FIG 6** The biofilm formation assay of seven CR-hvKP strains. The experiment was performed three times. The error bars indicate mean ± SD ($n = 3$).

facilitate homologous recombination between two plasmids, resulting in the formation of a novel fusion plasmid (41). Our study found multiple insertion elements, including IS*Kpn6*, IS*Kpn27*, IS*Yps3*, and IS*26*, in the vicinity of the *bla*$_{KPC-2}$ gene within the seven examined resistance plasmids. The presence of these MGEs suggests that recombination, insertion, or deletion events occur during the transmission of DNA fragments containing *bla*$_{KPC-2}$ and/or MDR regions between different strains. Interestingly, the sequence length of pCR-hvKP128-KPC-P1 and pCR-hvKP132-KPC-P1 was 154,719 bp, and the NCBI-Nucleotide BLAST comparison results showed that the coverage and identity of the two plasmids were 100%. This finding suggests that strains CR-hvKP128 and CR-hvKP132 might have originated from an ST11 MDR-KP/MDR-hvKP strain that horizontally transferred its MDR plasmid to other hvKP strains during nosocomial infection. Therefore, one of the significant evolutionary mechanisms of ST11 CR-hvKP is the transfer of MDR plasmids into ST11 hvKP strains facilitated by complete conjugative

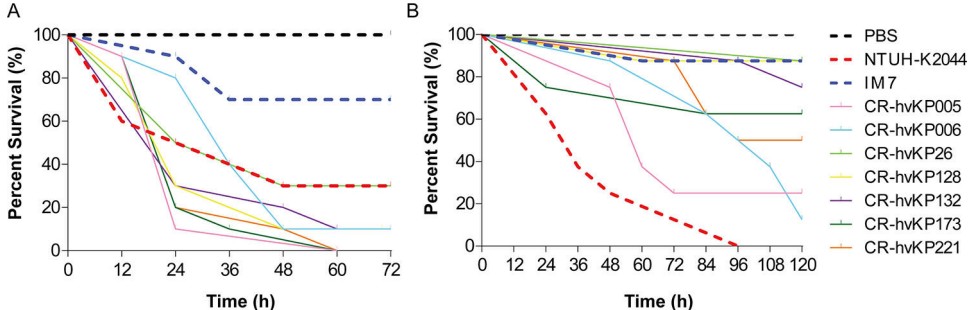

**FIG 7** Virulence characteristics of seven CR-hvKP strains *in vivo*. (A) Survival curves in the *G. mellonella* infection model. Groups of 10 *G. mellonella* larvae were subjected to injection of $2 \times 10^5$ CFU of each group of seven CR-hvKP strains, respectively. (B) Survival curves in the mice infection model. Groups of eight mice were infected with $5.0 \times 10^6$ CFU of different *K. pneumoniae* strains intraperitoneally. For A and B, NUTH-K2044 was used as a positive control, and IM7 was used as a negative control in these studies. The blank control group was injected with equal PBS. The experiment was performed three times.

transfer elements or conjugative plasmids. This hypothesis is supported by the findings from previous works (42, 43).

Fusion plasmid acquisition represents another significant pathway for the evolution of CR-hvKP (44). Xie et al. identified p17ZR-91-Vir-KPC, a hybrid plasmid, was formed by the fusion of a pLVPK-like plasmid and a $bla_{KPC-2}$-bearing plasmid in an ST11 CR-hvKP strain (45). Our study revealed a pLVPK-like plasmid named pCR-hvKP173-Vir-P1, which carried virulence genes and the multidrug efflux pump-related gene *oqxB*. Previous investigations have revealed that the *oqxB* gene could undergo horizontal transfer in certain Gram-negative bacterial pathogens, resulting in an increased antibiotic resistance rate (46). We hypothesize that the *oqxB* gene is horizontally transferred and fused with a pLVPK-like plasmid, forming pCR-hvKP173-Vir-P1. In addition to pCR-hvKP173-Vir-P1, we also identified another fusion plasmid, pCR-hvKP221-Vir-P2, which contained two virulence gene enrichment regions, a T4SS region, and an ARG enrichment region. This plasmid exhibits an oriT transfer origin and the transfer-related gene *traX/M*, suggesting a strong potential for conjugative transfer. Notably, the *G. mellonella* infection model demonstrated that CR-hvKP173 and CR-hvKP221 displayed higher virulence than the clinical cKP strain IM7. Intriguingly, pCR-hvKP221-Vir-P2 showed a high similarity to the *Escherichia coli* strain D6 plasmid A from an *Escherichia coli* strain, implying the potential spread of the virulence plasmid between different bacterial species and its strong pathogenicity. Furthermore, biofilm formation experiments indicated that CR-hvKP221 exhibited a high ability to form biofilms, indicating enhanced adhesion and antibiotic tolerance. Within CR-hvKP221, we also detected another pLVKP-like plasmid named pCR-hvKP221-Vir-P1. *K. pneumoniae* strains carrying pLVPK-like plasmids have been extensively disseminated in China, facilitating the integration of CR-hvKP resistance and virulence phenotypes (44, 47, 48). Our research indicates that the pLVPK-like plasmid can evolve into a fusion plasmid. This fusion plasmid can simultaneously carry resistance and virulence genes while also exhibiting a certain level of transmissibility within the local region. In our study, all CR-hvKP strains carried the *rmpA* gene and were positive in the string test, which may be due to the regulatory role of *rmpA* for hypermucoviscous phenotype (49). Yu et al. (50) found that the strain carrying *rmpA* was significantly associated with hypermucoviscous phenotype and pyogenic infectious diseases.

Two limitations of our study should be acknowledged. First, the co-transfer ability and mechanism of virulence plasmids and the $bla_{KPC-2}$-carrying plasmids in *Escherichia coli* and different types of *K. pneumoniae* were not investigated. The second is the small sample size of this study, which hinders us from investigating the prevalence of multiple highly homologous plasmids and the molecular evolution and transmission mechanism of these plasmids.

In conclusion, with seven ST11-K1 CR-hvKP strains collected from Hohhot during the past 4 years, we found that ST11-K1 CR-hvKP carried a similar plasmid containing the $bla_{KPC-2}$. These homologous plasmids have been observed for an extended period in the region. This finding provides valuable insight into the evolutionary mechanisms of ST11 CR-hvKP.

## MATERIALS AND METHODS

### Bacterial isolation and identification

We collected 124 isolates of CRKP between August 2015 and August 2020 at the Affiliated Hospital of Inner Mongolia Medical University (23). CRKP isolates were defined as those resistant to imipenem, meropenem, or ertapenem, as previously reported (19). The hypermucoviscous phenotype was determined by a string test, as previously described (51). The string test was positive if a single colony produced a viscous string longer than 5 mm. CR-hvKP was defined as a CRKP strain with a positive string test.

## Antimicrobial susceptibility

The antimicrobial agents assessed included aztreonam, amikacin, chloramphenicol, fosfomycin, tobramycin, ceftazidime, cefotaxime, meropenem, imipenem, gentamicin, tigecycline, ertapenem, ceftriaxone, ciprofloxacin, ceftazidime/avibactam, and polymyxin B. The minimum inhibitory concentrations (MICs) of these agents were determined using the agar dilution method (23). The antimicrobial susceptibilities were interpreted according to the guideline document M100-S31 established by the Clinical and Laboratory Standards Institute (CLSI). Tigecycline susceptibilities were determined following the criteria released by the U.S. Food and Drug Administration (FDA). *Escherichia coli* ATCC 25922 was used as the quality control strain throughout the study.

## Biofilm formation assay

The crystal violet staining method was used to evaluate the biofilm formation capability of CR-hvKP (52). In brief, 200 µL of mid-log phase bacteria ($1 \times 10^8$ CFU/mL), NTUH-K2044 isolate (positive control), and PBS (negative control) were added to 96-well microtiter plates and incubated overnight. Next, the cultures were removed, and the well was washed three times with phosphate-buffered saline (PBS). We used 1% crystal violet solution to stain for 20 minutes, followed by three additional washes. The solubilized stain was mixed with 200 µL of 95% ethanol, and the $OD_{600}$ value was measured. Each assay was conducted in duplicate and independently repeated three times. The stains with $OD_{600}$ values more than two times the PBS $OD_{600}$ value were considered strong biofilm-forming strains (53).

## *G. mellonella* infection model

Pathogen-free *G. mellonella* larvae weighing approximately 200–300 mg (Huiyude Biotech Company, Tianjin, China) were utilized to investigate the *in vivo* virulence of seven CR-hvKP strains, with 10 larvae assigned to each group. The hvKP strain NTUH-K2044 was a positive control for high virulence, and a clinical strain IM7 was used as a low virulence control. IM7 is a cKP strain with a negative string test isolated from the sputum samples of a 64-year-old patient admitted to the Affiliated Hospital of Inner Mongolia Medical University in 2016 with chronic obstructive pulmonary disease. The tested strains were diluted with PBS at $1 \times 10^8$ CFU/mL, and 2 µL of bacterial suspension was used to infect *G. mellonella* larvae. The infected larvae were then cultured at 37°C for 72 hours, and the survival rates were monitored. The negative control larvae received 2 µL of $1 \times$ PBS. Each assay was duplicated and independently repeated three times (54).

## Murine infection assay

The murine infection assay was conducted following the protocol described by Liao et al. (55). Female pathogen-free BALB/c mice (4–5 weeks old, weighing 15–20 g) from Beijing Vital River Laboratory Animal Technology Co., Ltd. were used in the experiment, with eight mice assigned to each group. The intraperitoneal injection was performed with 250 µL ($5.0 \times 10^6$ CFU) of the bacterial suspension. The mortality rate of the test mice was observed for 5 days after injection. The animal experiments were repeated twice to ensure data consistency (55, 56). The negative control (PBS), high virulence control (NTUH-K2044), and low virulence control (IM7) were included in the study.

## Conjugation assay

According to the methods described in previous studies (43, 55), we used the CR-hvKP test strains and rifampicin-resistant EC600 as the donor and recipient strains, respectively. They were cultured in Luria-Bertani (LB) broth until the log phase ($OD_{600}$ = 0.5–0.6). The EC600 and CR-hvKP cultures were mixed at a 3:1 ratio, and the mixture was plated on agar plates containing 5% sheep blood. After an overnight incubation, the putative transconjugants were resuspended and serially diluted in PBS. Each

diluted transconjugant was plated on MacConkey agar supplemented with rifampin (300 µg/mL) and MEM (4 µg/mL). A matrix-assisted laser desorption ionization time-of-flight mass spectrometry (MALDI-TOF MS) analysis was performed to identify the screened conjugated EC600 strain (57), and a PCR assay was used to validate the carbapenemase gene.

## Whole-genome sequencing and bioinformatics analysis

We used the Invitrogen PureLink Genomic DNA kit to extract Genomic DNA. The extracted genomic DNA was then subjected to sequencing using the Illumina HiSeq platform (Illumina, San Diego, CA, USA) and the Oxford Nanopore Technologies (ONT). Sequencing libraries were constructed using the TruSeq Nano DNA Sample Prep Kit (Illumina). Before assembling, the raw data underwent trimming and filtering procedures. This process includes removing adapter sequences, trimming low-quality reads (defined as reads with a quality score below Q20), eliminating sequences containing more than 10% ambiguous N bases, and excluding sequences shorter than 75 bp. The FastQC v0.11.3 was used to assess the quality of the reads (58).

The plasmid sequence is completed by *de novo* assembly using Unicycler (59). The Illumina data were used to evaluate the complexity of the genome and correct the Nanopore long reads. The genome was assembled by ABySS (http://www.bcgsc.ca/platform/bioinfo/software/abyss) with multiple-Kmer parameters. The marbl/canu (https://github.com/marbl/canu) was used to construct the Nanopore corrected long reads. GapCloser software was applied to fill the remaining local inner gaps and correct the single base polymorphism (https://sourceforge.net/projects/soapdenovo2/files/GapCloser/) for the final assembly results. The genome assembly statistics, including the number of scaffolds, median coverage of the scaffolds, predicted genome size, the N50 and N90 values, mean GC values and the number of circularised replicons, were observed to determine whether the assembly results were qualified. The complete circle of the genome was drawn with Circos v0.64 (http://circos.ca/). The circular plasmid was verified by the Circulator (http://sanger-pathogens.github.io/circlator/). The BLAST Ring Image Generator (BRIG) (60) and Easyfig (http://mjsull.github.io/Easyfig/) tools were employed to compare plasmid sequences.

The genomic sequences were then annotated using the NCBI Prokaryotic Genome Annotation Pipeline (PGAP). To confirm resistance genes, mutations in resistance genes, virulence factors, and metal resistance genes, the assembled contigs were subjected to BLAST analysis against the drug-resistant gene database (ResFinder) (http://genepi.food.dtu.dk/resfinder) (61), the virulence factor database (VFDB) (http://www.mgc.ac.cn/VFs/) (62), and the metal resistance genes database (BacMet) (http://bacmet.biomedicine.gu.se) (63). Plasmid replicon types were determined using the PlasmidFinder (64), and insertion sequences were identified using the ISFinder (65). The MLST 2.0 web tool (https://cge.food.dtu.dk/services/MLST/), provided by the CGE, was used to identify the sequence types (STs). The genome circle was visualized using the R.

## Phylogenetic analysis

A phylogenetic tree was constructed with 402 isolates, including seven CR-hvKP isolates obtained in this study and 395 references to CR-hvKP strains downloaded from NCBI in December 2022. The reference strains were selected with the following criteria: (i) they belonged to *K. pneumoniae* strains; (ii) they were representative of the major global problem clones, including eight MDR clones (CG15, CG20, CG29, CG37, CG147, CG101, CG258, and CG307) and six hypervirulent clones (CG23, CG25, CG65, CG66, CG86, and CG380) (66); (iii) they carried at least one of the following carbapenem resistance genes: *bla*KPC, *bla*NDM, *bla*VIM, *bla*IMP, and *bla*OXA-48; and (iv) they carried *rmpA*, *rmpA2*, *iroB*, *iucA*, and *peg-344* (67).

The wgSNPs identified through reference-based mapping for the CR-hvKP221 strains were used to build phylogenies using maximum likelihood (ML) as previously described (68). The MUMmer alignment software was used to find the different sites and

perform preliminary filtering by comparing the CR-hvKP genome sequence with the reference sequence. The Gubbins software (69) was used to identify recombinant regions and extract core genome SNPs from 402 CR-hvKP isolates. Finally, the MEGA software constructed a phylogenetic tree (70). The resulting tree was visualized using the Chi-plots (https://www.chiplot.online/) and Interactive Tree Of Life (iTOL) (https://itol.embl.de/) online software.

## ACKNOWLEDGMENTS

We are grateful to Dr. Jin-Town Wang from the Department of Microbiology, National Taiwan University, College of Medicine, for generously providing us with the NTUH-K2044 strain. We are very grateful to Dr. Fangyou Yu from the Department of Clinical Laboratory, Shanghai Pulmonary Hospital, School of Medicine, Tongji University, for generously providing us with the EC600 strain and technical support. We also appreciate the technical support of bioinformation analysis by Dr. Xi Yang from the Chinese Center for Disease Control and Prevention.

## AUTHOR AFFILIATIONS

[1]Department of Laboratory Medicine, The Affiliated Hospital of Inner Mongolia Medical University, Hohhot, China
[2]Department of Parasitology, The Basic Medical College of Inner Mongolia Medical University, Hohhot, China
[3]Medical Research Center, Beijing Institute of Respiratory Medicine and Beijing Chao-Yang Hospital, Capital Medical University, Beijing, China
[4]Department of Medical Experiment Center, The Basic Medical Sciences College of Inner Mongolia Medical University, Hohhot, China

## AUTHOR ORCIDs

Yu-Ling Han ⓘ http://orcid.org/0009-0005-5239-3074
Hua Wang ⓘ http://orcid.org/0000-0002-0746-0523
Zhi-De Hu ⓘ http://orcid.org/0000-0003-3679-4992
Wen-Qi Zheng ⓘ http://orcid.org/0000-0002-2970-3768

## FUNDING

| Funder | Grant(s) | Author(s) |
|---|---|---|
| School Science and Technology Team Project of Inner Mongolia Medcal University | YKD2022TD026 | Wen-Qi Zheng |
| General Project of Inner Mongolia Medical University | YKD2021MS011 | Wen-Qi Zheng |
| Inner Mongolia "Prairie Elite" Project Youth Innovation and Entrepreneurship Talent Training Program | ZY0130013 | Wen-Qi Zheng |

## AUTHOR CONTRIBUTIONS

Yu-Ling Han, Data curation, Formal analysis, Investigation, Methodology, Visualization, Writing – original draft | Hua Wang, Data curation, Formal analysis, Investigation, Methodology, Software, Visualization, Writing – original draft | Hong-Zhe Zhu, Methodology, Resources | Ying-Ying Lv, Resources | Wen Zhao, Supervision, Validation | Yan-Yan Wang, Resources, Validation | Jian-Xun Wen, Formal analysis, Methodology | Zhi-De Hu, Formal analysis, Methodology | Wen-Qi Zheng, Conceptualization, Methodology, Project administration.

## DATA AVAILABILITY

The genome sequences of seven CR-hvKP isolates were deposited in the GenBank database under the following accession numbers: CR-hvKP005, CP119013–CP119016; CR-hvKP006, CP119017–CP119020; CR-hvKP26, CP119021–CP119025; CR-hvKP128, CP119026–CP119028; CR-hvKP132, CP119029–CP119033; CR-hvKP173, CP119034–CP119039; and CR-hvKP221, CP119040–CP119046. The data of whole-genome sequences were deposited at the NCBI Sequence Read Archive (SRA) with the BioProject accession number PRJNA905913.

## ETHICS APPROVAL

This study was conducted following the ethical regulations of the Affiliated Hospital of Inner Mongolia Medical University.

## ADDITIONAL FILES

The following material is available online.

### Supplemental Material

**Captions (mSystems01101-24-s0001.docx).** Supplemental table captions.
**Table S1 (mSystems01101-24-s0002.docx).** The assembly results of seven CR-hvKP strains by Unicycler.
**Table S2 (mSystems01101-24-s0003.docx).** BLAST searching of the NCBI nucleotide database of the plasmids assembled in CR-hvKP221, CR-hvKP005, CR-hvKP006, CR-hvKP26, CR-hvKP128, CR-hvKP132, and CR-hvKP173 strains.
**Table S3 (mSystems01101-24-s0004.docx).** The function of resistance genes and virulence genes detected in CR-hvKP221, CR-hvKP005, CR-hvKP006, CR-hvKP26, CR-hvKP128, CR-hvKP132, and CR-hvKP173 strains.
**Table S4 (mSystems01101-2-s0005.docx).** SNP loci detected in 402 genomes using the MUMmer alignment software.

### Open Peer Review

**PEER REVIEW HISTORY (review-history.pdf).** An accounting of the reviewer comments and feedback.

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
