## [Reviewer comments · mSystems]

Phenotypic and genomic characterization of ST11-K1 CR-hvKP with highly homologous *bla*_{KPC-2}-bearing plasmids in China

Yu-Ling Han, Hua Wang, Hong-Zhe Zhu, Ying-Ying Lv, Wen Zhao, Yan-Yan Wang, Jian-Xun Wen, Zhi-De Hu, junrui wang, and Wen-Qi Zheng

Corresponding Author(s): Wen-Qi Zheng, The Affiliated Hospital of Inner Mongolia Medical University

Review Timeline:

Submission Date:	August 15, 2024
Editorial Decision:	September 30, 2024
Revision Received:	October 12, 2024
Accepted:	October 18, 2024

Editor: Anupama Khare

Reviewer(s): Disclosure of reviewer identity is with reference to reviewer comments included in decision letter(s). The following individuals involved in review of your submission have agreed to reveal their identity: Piklu Roy Chowdhury (Reviewer #1)

Transaction Report:

DOI: <https://doi.org/10.1128/msystems.01101-24>

Re: mSystems01101-24 (**Phenotypic and genomic characterization of ST11-K1 CR-hvKP with highly homologous *bla*_{KPC-2}-bearing plasmids in China**)

Dear Dr. Wen-Qi Zheng:

Thank you for the privilege of reviewing your work. As you can see below, Reviewer # 1 was mostly satisfied with the modifications you made to the manuscript, but had a few minor concerns. Below you will find my comments, instructions from the mSystems editorial office, and the reviewer comments.

Please address the remaining concerns of the reviewer. Additionally, the Genbank accession numbers for the assembled sequences have been mentioned. But has the raw sequencing data been deposited in an online database (e.g. SRA)? Once it has been deposited, please add the Accession number for the database to your Data Availability statement.

Please return the manuscript within 30 days; if you cannot complete the modification within this time period, please contact me. If you do not wish to modify the manuscript and prefer to submit it to another journal, notify me immediately so that the manuscript may be formally withdrawn from consideration by mSystems.

Revision Guidelines

Sincerely,
Anupama Khare
Editor
mSystems

Reviewer #1 (Comments for the Author):

The revised version of the manuscript is near complete, but it needs to be edited carefully for English. In addition, some of the

details inquired in my last review should be incorporated in the methods section and not left as responses in the rebuttal letter. I have highlighted them in my comments below.

PLEASE NOTE: Line numbers indicated below are from the "MARKED-UP" file.

1. Lines 112-113: I believe you mean the MIC of most antibiotics tested was over 128 μ g/ml? Please edit English.
2. Lines 120-124: Edit for English. This can be reduced to 2 sentences, with a reference to Figure S1 and Table S1.
3. Lines 139-140: You mean "over 70% plasmids contained" the genes? Edit sentence.
4. Line 143: Gene name starts with a lowercase "oqxB".
5. Line 164: GC content "ranges," not ranging.
6. Line 175: "the genetic context of," not "the genetic environment."
7. Lines 225-226: Gubbins is a "SNP-based core genome phylogeny" algorithm, so please mention that in the section.
8. Lines 229-231: Based on the concentric ring legends of Figure 5, isolates from China appear to be distributed all over the phylogenetic tree. However, authors appear to be highlighting one specific sub-clade in the tree in this section. Please color the subclade with a specific color or shade the sub-clade-both are easily available options in iTOL.
9. Figure 5 and the phylogeny section: Please turn the bootstrap value option or the "clade confidence score option in iTOL. If genealogical relationships are the focus of this section, the tree needs to have the values instead of branch lengths.
10. Lines 467-469 still do not mention that Trimmomatic was used to trim sequence and FastQC was used to assess the quality of the reads. These need to be incorporated in the text, not just the rebuttal letter!
11. Lines 482-484: Unicycler is a de-novo hybrid assembler; and when available, it uses both long reads and short reads to assemble a genome without a reference genome. The pipeline includes the circularization step for chromosomes and plasmids; therefore, the information in these lines is redundant. My comment in the last review was specifically to encourage the authors to use the 'assembly log' file and identify whether the scaffolds they have identified as plasmids were noted as "circularized" by Unicycler. The information is available in the 'assembly log' file generated by Unicycler. Therefore, without going into the details presented in these lines, just check the log file and state the facts.

Point-to-point response to reviewers

We are very thanks the reviewer for the constructive comments. My team has carefully read the reviewers' comments and revised the manuscript accordingly. All the amendments are highlighted in red in a "Marked Up Manuscript" PDF file. I hope the manuscript meets the publication criteria for *mSystems*.

This mail's end is the point-to-point response to the reviewers' comments.

Best wishes.

Wen-Qi Zheng, MD, on behalf of all authors.

Department of Laboratory Medicine, the Affiliated Hospital of Inner Mongolia Medical University, Hohhot, China.

Tel: 0471-3451230

E-mail: zhengwenqi2011@163.com

1. Lines 112-113: I believe you mean the MIC of most antibiotics tested was over 128µg/ml? Please edit English.

Response: Thank you for your suggestion. We have revised the sentence for clarity (lines 98–99).

2. Lines 120-124: Edit for English. This can be reduced to 2 sentences, with a reference to Figure S1 and Table S1.

Response: Thanks for your comments. Based on your previous suggestions, we have simplified the language in Tables S1 and S2 and added necessary data regarding the genome assembly process (lines 105–107). Genome assembly statistics have been uploaded as Table S1. Figure S1 and its legend have been deleted in the last revision. We have updated the query length and alignment length of the BLAST search in the updated Table S2 (line 107).

3. Lines 139-140: You mean "over 70% plasmids contained" the genes? Edit sentence.

Response: Thanks for your comments, the "over 70% plasmids contained" in lines 139-140 were contents in the last revised manuscript (ID: mSystems00579-24). Based on your previous suggestions, we have revised the description of the detection of virulence genes and antibiotic resistance genes in the manuscript, and we have corrected any inaccuracies (lines 116 - 121 and 129 - 131).

4. Line 143: Gene name starts with a lowercase "oqxB".

Response: Thank you for your reminder. We have standardized the representation of "oqx B " to lowercase throughout the manuscript (lines 117-121, line 162, lines 293-295).

5. Line 164: GC content "ranges," not ranging.

Response: Thank you for your reminder. We corrected it to "ranges" (line 142).

6. Line 175: "the genetic context of," not "the genetic environment."

Response: Thank you for your reminder. We have revised it to “context” (line 149).

7. Lines 225-226: Gubbins is a "SNP-based core genome phylogeny" algorithm, so please mention that in the section.

Response: Thank you for your suggestion. We have included a clarification in lines 196 - 197 regarding our use of Gubbins software. We have revised the materials and methods (lines 437 - 439).

8. Lines 229-231: Based on the concentric ring legends of Figure 5, isolates from China appear to be distributed all over the phylogenetic tree. However, authors appear to be highlighting one specific sub-clade in the tree in this section. Please color the subclade with a specific color or shade the sub-clade-both are easily available options in iTOL.

Response: Thank you for your suggestion. We have emphasized this unique sub-branch from China in Figure 5. We have enhanced the thickness of the color blocks (green) representing the Chinese region on the heat map.

9. Figure 5 and the phylogeny section: Please turn the bootstrap value option or the "clade confidence score option in iTOL. If genealogical relationships are the focus of this section, the tree needs to have the values instead of branch lengths.

Response: Thank you for your reminder. In accordance with your previous suggestions, we have marked the bootstrap values in Figure 5 using red font and provided comprehensive explanations in the caption. Additionally, we have revised the tree to display bootstrap values instead of branch lengths in lines 206–208.

10. Lines 467-469 still do not mention that Trimmomatic was used to trim sequence and FastQC was used to assess the quality of the reads. These need to be incorporated in the text, not just the rebuttal letter!

Response: Thank you for your reminder. “The FastQC v0.11.3 was used to assess the quality of the reads” was modified in line 399. Trimmomatic software was not used in

this study. In accordance with your previous suggestions, We have modified the method and material in the revision (line 399). Genome assembly statistics including the N50 and N90 values, GC content (%) values, Number of scaffolds, and the number of circularised replicons have been uploaded to Table S1 in the revision.

11. Lines 482-484: Unicycler is a de-novo hybrid assembler; and when available, it uses both long reads and short reads to assemble a genome without a reference genome. The pipeline includes the circularization step for chromosomes and plasmids; therefore, the information in these lines is redundant. My comment in the last review was specifically to encourage the authors to use the 'assembly log' file and identify whether the scaffolds they have identified as plasmids were noted as "circularized" by Unicycler. The information is available in the 'assembly log' file generated by Unicycler. Therefore, without going into the details presented in these lines, just check the log file and state the facts.

Response: Thank you for your professional advice. Based on your previous suggestions, we have eliminated this section and incorporated references. The genomes and plasmids involved in our manuscript are assembled with reference to the method of this article (PLoS Comput Biol. 2017 Jun 8;13(6):e1005595) (line 399). Genome assembly statistics have been uploaded as Table S1 in the last revision. The verification software for circular plasmid is circulator (<http://sanger-pathogens.github.io/circlator/>), and we have modified the method and material (lines 400 - 409).

Additionally, to provide readers with a concise overview of key plasmids (the resistant and virulent plasmids), we have included the molecular characterization of the chromosome, resistant, and virulent plasmids of the CR-hvKP isolates in Table 3. The molecular characterization of other small plasmids and their BLAST results compared to similar reported plasmids have been organized into Table S2.

Re: mSystems01101-24R1 (**Phenotypic and genomic characterization of ST11-K1 CR-hvKP with highly homologous *bla*_{KPC-2}-bearing plasmids in China**)

Dear Dr. Wen-Qi Zheng:

Your manuscript has been accepted, and I am forwarding it to the ASM production staff for publication. Your paper will first be checked to make sure all elements meet the technical requirements. ASM staff will contact you if anything needs to be revised before copyediting and production can begin. Otherwise, you will be notified when your proofs are ready to be viewed.

Sincerely,
Anupama Khare